# Exceptionally Fast Temperature-Responsive, Mechanically Strong and Extensible Monolithic Non-Porous Hydrogels: Poly(*N*-isopropylacrylamide) Intercalated with Hydroxypropyl Methylcellulose

**DOI:** 10.3390/gels9120926

**Published:** 2023-11-24

**Authors:** Beata Strachota, Adam Strachota, Leana Vratović, Ewa Pavlova, Miroslav Šlouf, Samir Kamel, Věra Cimrová

**Affiliations:** 1Institute of Macromolecular Chemistry, Academy of Sciences of the Czech Republic, Heyrovskeho nam. 2, 162 00 Praha, Czech Republic; beata@imc.cas.cz (B.S.); vratovic@imc.cas.cz (L.V.); pavlova@imc.cas.cz (E.P.); slouf@imc.cas.cz (M.Š.); cimrova@imc.cas.cz (V.C.); 2Cellulose and Paper Department, National Research Centre, 33, El-Bohouth Str., Dokki, Giza 12622, Egypt; samirki@yahoo.com

**Keywords:** hydrogels, drug release, smart materials, poly(*N*-isopropylacrylamide), cellulose, semi-interpenetrating networks

## Abstract

Exceptionally fast temperature-responsive, mechanically strong, tough and extensible monolithic non-porous hydrogels were synthesized. They are based on divinyl-crosslinked poly(N-isopropyl-acrylamide) (PNIPAm) intercalated by hydroxypropyl methylcellulose (HPMC). HPMC was largely extracted after polymerization, thus yielding a ‘template-modified’ PNIPAm network intercalated with a modest residue of HPMC. High contents of divinyl crosslinker and of HPMC caused a varying degree of micro-phase-separation in some products, but without detriment to mechanical or tensile properties. After extraction of non-fixed HPMC, the micro-phase-separated products combine superior mechanical properties with ultra-fast T-response (in 30 s). Their PNIPAm network was highly regular and extensible (intercalation effect), toughened by hydrogen bonds to HPMC, and interpenetrated by a network of nano-channels (left behind by extracted HPMC), which ensured the water transport rates needed for ultra-fast deswelling. Moreover, the T-response rate could be widely tuned by the degree of heterogeneity during synthesis. The fastest-responsive among our hydrogels could be of practical interest as soft actuators with very good mechanical properties (soft robotics), while the slower ones offer applications in drug delivery systems (as tested on the example of Theophylline), or in related biomedical engineering applications.

## 1. Introduction

Hydrogels possess exceptional properties such as a high content of water, softness, flexibility and biocompatibility [1]. For these reasons, they have found numerous applications in medicine and in tissue engineering [2]. The so-called smart hydrogels, to which the materials studied in this work belong, are capable of responding to external stimuli by volume changes or by force [3].

The most popular temperature-sensitive polymer is poly(*N*-isopropylacrylamide) (PNIPAm) [4]. Hydrogels based on this polymer show a distinct ‘volume-phase transition’ at temperatures above 32 °C [5], at which point the polymer becomes considerably less hydrophilic, which leads to deswelling of crosslinked water-swollen PNIPAm.

The rate of stimulus-response is of crucial importance for potential applications of PNIPAm hydrogels: In some cases (e.g., drug release), a moderate rate is needed, while in others, such as mechanical actuators, the fastest possible one is desirable. The possible long-distance-acting mechanisms of water expulsion during shrinking, and of water uptake during re-swelling decide about the rate of temperature-response.

A need for optimization concerns the poor mechanical properties, which still present a major limitation of the practical use of thermo-sensitive hydrogels, such like most of the PNIPAm-based ones. A variety of approaches have been tested to improve the mechanical properties [6], including the synthesis of a highly regular and homogeneous network structure [7], the introduction of ionic groups into the polymer network [8], the use of polyrotaxane derivatives as cross-linkers [9], or the synthesis of semi-interpenetrating [10], or interpenetrating [11] gel networks (‘double networks’). Finally, the incorporation of nanofillers generally can provide a tremendous reinforcement to polymer matrixes, including hydrogels (see e.g., [12]), due to the nanofillers’ high specific surface area.

Some nanofillers make possible the simultaneous improvement of both the mechanical properties and of the stimulus-response rate. One such example were the in-situ-formed silica nanospheres in nanocomposite PNIPAm hydrogels studied by the authors in their previous research [13,14,15]. Similar hydrogels also were obtained with nano-TiO_2_ [16]. Interesting results were obtained by the authors with PNIPAm/clay nanocomposite hydrogels [17,18,19], which were first developed by Haraguchi [20]. The latter gels displayed exceptional mechanical and tensile properties, but they were able of very fast temperature response only in the highly porous state, in which they were extensible but very soft [19,21].

Achieving fast responsiveness in non-porous centimeter-sized monolithic hydrogel samples is difficult due to the role of diffusion, as well as of the length of diffusion paths in the course of the process of expelling or taking-up of water. ‘Skin/core effects’ [22], which occur in monolithic 3D samples, further hinder the process. Solutions aimed at accelerating the stimulus-response rate of such hydrogels typically were based on introducing some structural heterogeneity in the non-porous materials. The most successful methods included the incorporation of dendrimer molecules (polyamidoamine [23]) as semi-interpenetrating network embedded in the PNIPAm network, or the synthesis of networks which contain brush structures as elastic chains (e.g., PNIPAm grafted on PNIPAm [24]). In both the latter works, response times of ca. 10 min for 70% of the shrinking were achieved.

Fast-responsive non-porous hydrogels also were obtained by the authors in their previous works, namely PNIPAm hydrogels reinforced with amylopectin starch [25,26]. The reinforcing phase was found to markedly improve the tensile properties of the gels, and at proper concentrations of the embedded starch phase, the rate of temperature-induced hydrogel shrinking was tremendously accelerated. Another non-porous bulk system, which displayed rapid shrinking response to temperature (and also to pH), and which was developed by the authors, was a PNIPAm/clay nanocomposite hydrogel doped by sodium methacrylate co-monomer [27]. Its rapid kinetics of stimulus-response was attributed to reversible micro-phase-separation in the gel, which was triggered by the T- and/or pH-stimulus. The present work is motivated by the authors’ interest in systems similar to the previously studied PNIPAm/starch nanocomposites, but intercalated by linear and well-soluble hydroxypropyl methylcellulose (HPMC) molecules.

HPMC is an easily water-soluble derivative of cellulose, that is physiologically harmless and hence is popular in pharmaceutical and foodstuff applications [28], as a matrix for controlled drug release [29], as well as for tissue engineering [30]. Cellulose derivatives like HPMC also were used in fast-responsive polysaccharide-based hydrogels. However, the vast majority of such systems is based on highly porous morphologies, which greatly enhance the rate of swelling or deswelling [31].

Chemically more complex are porous interpenetrating PNIPAm/polysaccharide (‘double’) networks (IPN), e.g., such consisting of crosslinked PNIPAm and crosslinked hydroxypropyl cellulose [32], which in the freeze-dried state were able of fairly fast re-swelling (80% in 10 min). Another example of highly porous double networks was based on IPN of PNIPAm and carboxymethyl cellulose, where the latter polymer was crosslinked by calcium cations [33]. This system was sensitive to both temperature and pH, and was tested for drug release (release times between 100 and 200 min).

Semi-interpenetrating (SIPN) highly porous PNIPAm-based networks, in which the temperature-sensitive PNIPAm network was intercalated by a linear or branched polysaccharide, also were studied and displayed similar practical properties like the above-discussed IPNs. Drug release studies were carried out with PNIPAm intercalated with linear HPMC [34] (which under the reaction conditions might have formed some covalent links to PNIPAm), achieving release times between 60 and 120 min. Neat microcrystalline cellulose also was employed as component of ‘semi-interpenetrating’ networks, which at the same time might be regarded as organic-organic nanocomposites, in which cellulose was embedded in the form of molecular rigid rods into a PNIPAm network [35]. After freeze-drying, the mentioned SIPN hydrogel (now porous) was tested for drug release. It displayed rapid deswelling (10 min), as well as fast re-swelling (25–60 min). Porous PNIPAm-based SIPN hydrogels were synthesized also with polysaccharides different from cellulose or starch, like salecan [36] or succinoglycan [37]. The porous PNIPAm/succinoglycan gels [37] displayed T-induced deswelling in 20 min, re-swelling in 16 h, and drug release in 6 to 12 h.

Polyacrylate polymers grafted onto polysaccharides also were investigated concerning their swelling properties in a few research works [38,39], which, however, always focused on freeze-dried porous products.

In contrast to their highly porous variety, in the non-porous PNIPAm/polysaccharide gels, the rapid stimulus response was much more difficult to achieve, as already mentioned further above. Nevertheless, in a few rare works, some promising materials have been obtained, including the monolithic non-porous SIPN sodium carboxymethyl cellulose/PNIPAm [40], in which the re-swelling time was markedly reduced by the filler phase, from more than 12 h down to 3 h. PNIPAm/chitosan gels [41] also were found to display improved kinetics of swelling-related processes like drug release (process duration: 50–250 min). Similar acceleration of stimulus-response was observed in PNIPAm/alginate SIPN hydrogels [42]. The gels were able of moderate (4 to 15%) volume responses to 5-min-pulses of T or pH, while complete shrinking was achieved in 50 to 100 min. The further-above-mentioned previous research of the authors into PNIPAm hydrogels reinforced with amylopectin starch also belonged to the difficult topic of fast-responsive non-porous PNIPAm/polysaccharide gels [25,26].

The aim of the present study was to develop mechanically strong and extensible monolitic non-porous hydrogels, which would be able of extensive and simultaneously very fast, or moderately fast, as well as tunable temperature-induced shrinking. This aim was to be achieved by the synthesis of a regular network, consisting of chemically crosslinked PNIPAm intercalated with hydrophilic linear HPMC macromolecules, a biodegradable and biocompatible nature-sourced polymer. The structure-property relationships in the intercalated gels based on different PNIPAm/HPMC ratios were to be elucidated. The analysis of the shear moduli of the products was performed in the same way like in [43].

In contrast to the previously developed promising PNIPAm/starch gels, in the studied PNIPAm/HPMC system the linear intercalated phase was to be monomolecularly dispersed in the PNIPAm network, due to the high solubility of HPMC in water.

The developed materials could be of interest for applications such as robust drug- or reagent release materials (if they possess moderately fast T-response), or as one-way actuators (very fast T-response). Drug release tests at physiological temperature were to be performed.

## 2. Results and Discussion

### 2.1. Synthesis

The studied intercalated hydrogels were prepared via free radical polymerization and crosslinking of *N*-isopropylacrylamide) (NIPAm) in an aqueous solution under argon atmosphere, in the presence of dissolved linear hydroxypropyl methyl cellulose (HPMC), as shown in Figure 1. The obtained type of structure is often called ‘semi-interpenetrating network’ in the literature, as mentioned in the Introduction. The synthesis details are given in the Experimental Part and also in Appendix A.

The *N*,*N′*-methylenebisacrylamide co-monomer (BAA) was employed as chemical crosslinker in PNIPAm, at concentrations of 1, 2, and 4 mol% of all double bonds. Higher crosslinker contents were not tested, because of the authors’ experience from previous work [15] which indicated a distinct and rapid decrease of maximum swelling degree, as well as of the gel shrinking ratio, with the increasing crosslinker content. The crosslinker content expectedly strongly influenced the mechanical properties, but also the tendency of HPMC to disperse homogeneously in PNIPAm (see morphology discussion further below).

HPMC was embedded as an intercalated phase, without performing any chemical coupling reaction to the forming PNIPAm. HPMC concentration in the synthesis mixture (and also in the final hydrogels in the ‘as prepared’ state) was 1, 2, 3, and 5 wt.%, respectively. For that purpose, the appropriate amounts of HPMC were dissolved in the planned amount of reaction solvent (water, see Appendix A) and stirred for 1 day prior to the synthesis itself, thus yielding HPMC solutions with concentrations of 1.3, 2.7, 4.1, and 6.9 wt.%, respectively, as reaction medium for the polymerizations. The last-mentioned among the HPMC solutions already was considerably viscous, which somewhat complicated the synthesis (i.e., the stirring of the subsequently added components). Also, the mechanical properties of the obtained hydrogels were observed to significantly drop, if going from 3 to 5 wt.% of HPMC in the synthesis mixture. Hence, higher HPMC contents were not tested. The HPMC contents in dry hydrogels were calculated to be: 10.6, 19.4, 26.8, and 38.3 wt.% (see also Appendix A).

The radical co-initiator system ammonium peroxodisulfate (APS)/*N*,*N*,*N′*,*N′*-tetramethylethylene diamine (TEMED) was chosen for the syntheses of the studied hydrogels (see Figure 1a), since it is active at room temperature and even at much lower temperatures (see e.g., [13]). Thanks to this, heating (or reaction-warming) above the LCST point of PNIPAm was possible to avoid, and with it an undesired PNIPAm precipitation.

The concentration of the NIPAm monomer in the reaction mixture always was kept equal to 0.75 mol/L, corresponding to ca. 8 wt.% (depending on mixture). This optimized concentration was chosen as standard in view of the authors’ previous experience with NIPAm polymerization by TEMED/APS [17,18]. The swelling degrees (*Q* = *m_swollen_*/*m_dry_*) of the ‘as prepared’ hydrogels, after taking out from the reaction tube, ranged between 12 (no HPMC) and 8 (with 5 wt.% of HPMC in the reaction mixture), as shown in Figure 1.

The final step of the preparation of the hydrogels, which was carried-out for most of the specimens, was their equilibrium swelling. After this treatment, the gels significantly increased their volumes, reaching swelling degrees of 26–32 in case of the gels containing 1 mol% of BAA crosslinker, in which the highest swelling was achieved at the highest HPMC content. *Q* = 19–27 was obtained with 2 mol% of BAA, and *Q* = 16–20 with 4 mol% of BAA. Some of the specimens were studied in the ‘as prepared’ state, without the swelling treatment. The swelling degrees in the as-prepared, and in the equilibrium-swollen state are shown in Figure 1.

The equilibrium swelling procedure simultaneously led to extraction of sol from the gels, which was desirable. Eventual unreacted monomers, as well as the extraction-able fraction of HPMC which was not permanently fixed in the PNIPAm network (see Figure 2), were removed.

Interestingly, it was found that HPMC was extracted to a great part from all the prepared hydrogels, and that only ca. 1–5 wt.% of the initial amount of HPMC stayed permanently embedded, leading to HPMC loadings of ca. 0.1 to 3 wt.% in dry final products (see also Appendix A). This was determined by the evaporation of the extracted sol, and by its gravimetric, as well as NMR analysis (Appendix A). The latter indicated that the sol was pure HPMC, without any NIPAm or PNIPAm. The gels’ outward appearance, which was transparent or turbid, did not change during the swelling/extraction process.

Some of the interesting properties of the studied gels, i.e., their fast or ultra-fast deswelling in response to temperature, their morphology, moduli, or their tensile properties, can be directly attributed to the specific internal structure after the HPMC extraction (nano- or sub-micrometer porosity, see Figure 2) and also to the presence of the residual, permanently fixed HPMC fraction (see Figure 2).

The described extraction behavior is in contrast to the results obtained for some PNIPAm–polysaccharide semi-IPN systems studied in the literature, in which the linear polysaccharides were reported to be fairly stable against washing-out [34,36]. In all cases, the small or large fixed fractions of linear polysaccharide likely are the result of chain-transfer or oxidation reactions on the polysaccharide chains during the radical polymerization of NIPAm. Such reactions probably were especially prominent in [34], due to the vigorous conditions of the persulfate-initiated polymerization at elevated temperature. In contrast to that, the low-temperature radical polymerization employed in the present work led to a modest fixation of the polysaccharide.

### 2.2. Morphology of the Prepared Hydrogels

The Figure 2 shows the outward appearance of the prepared bulk non-porous PNIPAm/HPMC hydrogels. The products display the same appearance (except the change in size) in the ‘as prepared’ state, as well as in the final state after equilibrium swelling. The swollen state is shown in Figure 2. It can be observed that the amount of the BAA crosslinker in the PNIPAm network has a very strong influence on the homogeneity (transparency) of the obtained PNIPAm/HPMC hydrogels, especially at higher HPMC contents.

With 1 mol% of BAA, all the hydrogels are very transparent and nearly identical in appearance. The effect of increasing amounts of HPMC is hardly visible, although in case of the sample 1B-5H containing 5 wt.% of HPMC during synthesis, some opalescence can be noted (faint bluish hue).

In more crosslinked systems, with 2 or 4 mol% BAA, phase separation of HPMC becomes prominent: In the series with 2 mol% of BAA, already the sample 2B-2H (2% HPMC during synthesis/19.4 wt.% in dry gel) displays notable turbidity and opalescence, while still being translucent. Only 2B-1H is fully transparent. With 4 mol% of BAA crosslinker, already the sample with the smallest tested HPMC content, 4B-1H is strongly opalescent and cloudy. In case of 4B-5H the micro-phase-separation is very pronounced: the gel is completely opaque and white.

The narrower mesh of the hydrogels with higher-crosslinked PNIPAm obviously does not favor an easy molecular dispersion of HPMC in them, so that an increasing fraction of the intercalated HPMC phase is embedded in the form of aggregates (domains), as illustrated in Figure 3.

After the swelling treatment, the submicronic optical heterogeneity persists, as the extracted HPMC domains leave behind submicro-cavities filled with a small amount of HPMC plus pure water (see Figure 2 further above). The cavities also display a difference in refraction index relatively to hydrated PNIPAm, similar to HPMC domains before extraction.

#### 2.2.1. Light Microscopy (LM)

Light microscopy was employed to investigate the turbid gels at the scale of tens of micrometers. Some representative LM images are shown in Figure 3a,b (‘as prepared’ hydrogels) and in Figure 3c–h (after equilibrium swelling). It can be seen that the opaque hydrogel 4B-2H displays practically the same granular pattern in the LM images taken in the ‘as prepared’ state (Figure 3b), and in the equilibrium-swollen (and extracted) state (Figure 3h). On the other hand, the transparent but slightly opalescent hydrogel 1B-5H (prepared with the maximum achievable HPMC content) appears completely homogeneous in the LM images, both in the ‘as prepared’ and in the ‘swollen + extracted’ state (see Figure 3a and Figure 3e, respectively). The strongly opalescent and cloudy sample 4B-1H (in the ‘swollen + extracted’ state) also does not display any easily visible heterogeneity in LM. The highly transparent HPMC-filled and HPMC-free gels expectedly display no patterns in LM (see samples 1B, 4B, and 1B-1H in Figure 3c, Figure 3f and Figure 3d, respectively).

#### 2.2.2. Transmission Electron Microscopy (TEM)

‘As prepared’ and equilibrium-swollen specimens were both investigated by TEM. With the help of staining by uranyl acetate, the phase structure of the highly opaque gels could be analyzed by this method (see Figure 4). It was observed that the contrast agent preferentially adsorbs onto the HPMC filler phase (dark fibrous patterns), which is attributed to the better chelating properties of HPMC repeat units (potential multi-dentate ligand), if compared to PNIPAm (monodentate ligand).

Slow drying of the studied samples in air was performed until a glassy consistence was achieved, after which drying in vacuum at 60 °C was applied (see Experimental Part). This was done in order to obtain realistic morphology results: The so-prepared specimens of the non-porous monolithic gels stayed non-porous also after drying, in contrast to eventual specimens which would be prepared by freeze-drying (a rapid method popular in the literature). It should be also noted, that the ‘gently dried’ glassy specimens are able of slow re-hydration (restoration of the original equilibrium swollen state), whose duration is approximately 3 days in case of the large specimens, while thin slices re-hydrate immediately, e.g., the micro-specimens cut for the TEM analyses.

The morphology of the highly opaque and HPMC-rich gel “2B-5H-as-prepared” (non-extracted, with nominally 38 wt.% of HPMC in the dry gel) is shown in the Figure 4a–c. The optical photograph of the same material is shown in Figure 4a: The material is non-porous, but a core-shell macro-structure can be observed, with a transparent outer layer (dark in Figure 4a), and a white opaque core part of the sample. The opacity (hence phase-separation) in the core region is caused by the mobility of the extractable HPMC phase, which during the slow drying of the sample diffuses into the still wet core part of the cylindrical specimen. HPMC is enriched there and eventually undergoes phase-separation. The enrichment of HPMC in the core region also was proven by FTIR (as documented in Appendix A). The TEM image of the HPMC-rich core of the sample 2B-5H-as-prepared (Figure 4b,c) displays a very dark interconnected pattern of fibrous structures. This pattern of stained HPMC is overlaid with the lighter grey pattern of the holey carbon foil (see exemplary miniature image of it below Figure 4e) which served as the sample substrate. The PNIPAm matrix is lighter grey.

The equilibrium swollen (and extracted) samples 2B-5H and 4B-5H display no macroscopic phase separation of HPMC after the slow drying (see exemplary optical photograph in Figure 4d). In contrast to the highly opaque swollen state (shown in Figure 2), both dried samples are highly transparent. This means that the submicronic cavities left behind by the extraction of HPMC collapsed and closed (or at least greatly shrunk) during the drying. In the TEM images of the extracted samples (Figure 4e–h), the pattern of HPMC is similar, but much fainter than in “2B-5H-as-prepared”. It can be noted that the pattern of residual HPMC is darker in the extracted sample 4B-5H, which was synthesized with the same HPMC loading like 2B-5H, but 4B-5H contained more crosslinker. These results clearly visualize the persisting amount of fixed HPMC in the hydrogels after the equilibrium swelling. Generally, it can be also noted, that in all the samples, in which HPMC was phase-separated, it is arranged into bent fibrous patterns.

### 2.3. Moduli of the Hydrogels

The prepared PNIPAm/HPMC hydrogels display moduli which are typical for simple, covalently crosslinked hydrogels. However, as will be demonstrated further below, these moduli are combined with tremendously improved tensile properties, as well as with fast or even ultra-fast stimulus-responsiveness. The moduli of the prepared PNIPAm/HPMC hydrogels were measured in compression experiments, both in the as-prepared- (“ap”) and in the equilibrium-swollen state (“sw”). The results are summarized in Figure 5 and Appendix A.

Higher amounts of the chemical crosslinker expectedly raise the moduli in most cases (see Figure 5 and some details in Appendix A). However, in samples with the highest loading of the intercalated HPMC phase (5%), the effect of increasing micro-phase separation at high crosslinking becomes dominant (see last group of columns in Figure 5a,b). In the “ap” state the modulus even decreases if going from low-crosslinked 1B-5H to 4B-5H.

The net effect of increasing HPMC loading on the moduli generally is moderate, as can be seen in Figure 5a,b (and some details in Appendix A). In the samples 4B-1H-ap, 4B-1H-sw, and 4B-2H-sw, a moderate net reinforcing effect can be seen. Otherwise, the moduli are rather unchanged, especially in the 1B and 2B series, until the loading of 5% of HPMC, where in the 2B and 4B series the effect of micro-phase-separation significantly reduces them. The effect of the HPMC polysaccharide chains on the moduli results from two mutually opposed contributions: First from hydrogen-bonding PNIPAm–HPMC (see Figure 4), and second, from network loosening (‘dilution effect’) via intercalation and eventual micro-phase-separation, which after extraction of the HPMC domains leads to nano-cavities in the “sw” state. In the most-crosslinked 4B series, both these contributions are most dramatic: The effect of H-bonding at first brings significant net reinforcement, while at higher HPMC loadings the micro-phase-separation leads to a sizeable drop in modulus, especially in the “ap” state. The high covalent crosslinking in the 4B series ‘multiplies’ the effect of H-bonding between the constituent phases (simple effect at low HPMC contents, see Figure 4b). The effects of network dilution and of H-bonding on modulus in PNIPAm/polysaccharide systems was previously studied by the authors on the PNIPAm/amylopectin system in [25,26] and a more detailed discussion is provided in the Appendix A.

In the “sw” state the trends in moduli are similar like in the “ap” state (see Figure 5b vs. Figure 5c), but the “sw” moduli are smaller. However, in case of the 2B series, the moduli are comparable in the “ap” and “sw” state, which highlights the importance of the hydrogen bonding shown in Figure 4. Its effect was evaluated by analyzing the changes in the nominal number of crosslinks per specimen (expressed as *r*_1_/*r*_2_). This was done by comparing moduli in the “ap” and in the “sw” state, in view of their respective swelling degrees. The relevant data are summarized in Table 1. The mathematic evaluation is explained in detail in the Appendix A. Its results are illustrated in Figure 5c, as plots of the magnitude *r*_1_/*r*_2_, which expresses the relative change in the number of crosslinks per specimen, vs. the samples composition. The value *r*_1_/*r*_2_ = 1 means no change in the number of crosslinks, whereas *r*_1_/*r*_2_ > 1 indicates an increase of crosslink amount in the “sw” state. On the other hand, *r*_1_/*r*_2_ < 1 means a decrease in the number of crosslinks in the “sw” state. It can be seen that the ratio *r*_1_/*r*_2_ is greater than one in nearly all the tested samples. The value of *r*_1_/*r*_2_ always distinctly increases with the content of HPMC present during the gels’ synthesis (see Figure 5c). This increase is especially marked in the 2B and 4B series. The *r*_1_/*r*_2_ values are the highest in the 2B family, in which the combination of its covalent crosslinking density with the still modest effects of micro-phase-separation leads to the maximum effect of H-bonding between PNIPAm and HPMC. A detailed discussion of the trends in moduli is provided in the Appendix A.

### 2.4. Tensile Properties

The tensile properties of the chemically crosslinked PNIPAm hydrogels are tremendously improved by the incorporation of HPMC (see Figure 6). Hydrogen bonding between HPMC and the PNIPAm matrix, as well as intercalation effects appear to play an important role both in the HPMC-rich ‘as prepared’ gels, as well as in the equilibrium-swollen ones, in which the HPMC content is low (0.1 to 3 wt.% in dry gels). Micro-phase-separation also affected the tensile properties, especially if it was very strong (with 5% HPMC and 4 mol% BAA), but even such gels displayed much better properties than their HPMC-free analogues. The tensile curves of the studied products are compared in Figure 6 and Appendix A, while the material characteristics obtained from the graphs, like toughness, strength and extensibility, are compared in Appendix A, and are also listed in Appendix A. The studied hydrogels additionally displayed a remarkable durability in ‘crushing tests’ conducted in the compression mode, surviving up to 80% compression (see Figure 7 and Appendix A).

The increasing amount of HPMC led to a higher extensibility and to higher toughness, except in case of the most strongly phase-separated sample 4B-5H-ap. The toughness trend was especially strong in the ‘as prepared’ gels. On the other hand, the ‘differential moduli’ (curve slopes) over extended ranges of deformations were dropping with increasing HPMC loading. The stress at break (tensile strength) displayed a moderate increase with HPMC loading in the less crosslinked gels, such as the 1B and 2B-series. In the 4B series, the stress at break increases until 2 wt.% of HPMC, after which it significantly drops. The described trends can be explained by the intercalation effect of HPMC. Its higher contents lead to looser and more extensible networks. Additionally, dynamic hydrogen bonds between PNIPAm and HPMC are responsible for the increase in strength, which however can be counteracted by micro-phase-separation in the HPMC-rich specimens of the 4B series.

The increasing amount of the BAA crosslinker expectedly led to higher differential moduli (slopes), to higher values of tensile strength, but also to smaller extensibility (see Figure 6 and Appendix A).

Micro-phase separation played an important role in the “ap” gels: Its effect is clearly visible on the example of the highly crosslinked and highly filled sample 4B-5H-ap, where this phase separation causes reduced extensibility (crack initiation; see Figure 6b), as well as a marked drop in toughness and strength (see Figure 6b and also Appendix A). Besides this, the micro-phase separation contributes to the softening of the gels with increasing HPMC content (see long-range differential moduli in Figure 6b).

Interestingly, the hydrogels after swelling display comparable tensile properties like the ones in the ‘as prepared’ state, as demonstrated by the comparison of the series 4B-ap and 4B-sw (see Figure 6b,c and Appendix A). The trends in the swollen gels are similar but simpler than in the as-prepared ones: The extensibility systematically increases with the content of HPMC present during synthesis, also in case of 4B-5H-sw: The removal of the excessive phase-separated HPMC cancels its crack-initiating effect, and there remains only the effect of intercalation, which leads to a looser and hence more extensible network. In comparison with the ‘as prepared’ gels, the extensibility is moderately reduced in the swollen ones (except in the mentioned composition 4B-5H-sw), which can be attributed to the more stretched elastic chains in the swollen state. For the same reason, the differential moduli in the swollen gels always are higher than in the “ap” state. In the swollen state, the tensile strength of the HPMC-intercalated samples (in the 4B series) is much higher than that of the neat 4B matrix, but the strength values steadily decrease with increasing content of HPMC present during the synthesis. Moreover, the strength values of the “sw” gels are either significantly higher (4B-1H-sw) or practically equal (4B-3H-sw and 4B-5H-sw) like in the “ap” state of the same samples. The decreasing trend of the strength can be explained by an increasingly looser structure with nano-voids in the place of the extracted HPMC. The toughness of the swollen gels of the 4B series (except the HPMC-free 4B-sw) is fairly high: it is significantly but not dramatically surpassed only by two of the toughest ‘as-prepared’ samples among the gels tested, namely by 4B-3H-ap and 1B-5H-ap. The toughness of the swollen 4B gels slightly increases with the content of HPMC present during the synthesis, which could be attributed to the effect of the increasing amount of the permanently fixed HPMC (see Figure 7).

### 2.5. Temperature Response

A very valuable ‘smart’ property of the studied PNIPAm/HPMC hydrogels is their extensive volume response to temperature changes, which in some cases proceeds with an ultra-fast rate in spite of the gels’ non-porosity and large size. The thermo-sensitive swelling behavior was investigated on specimens which previously underwent equilibrium swelling at room temperature. Such specimens did not change their composition during the experiments. The curves of the temperature-dependent swelling degrees (*Q*) of the 1B and 4B series are shown in Figure 8, while the similar results for the 2B series can be found in Appendix A.

All the studied PNIPAm/HPMC hydrogels exhibit a dramatic *T*-induced drop in swelling degree (*Q*), which is the steepest upon esceeding 32–33 °C, as illustrated by Figure 8. This LCST point is identical in all hydrogels, because they all consist of nearly pure PNIPAm (the highest HPMC content was ca 3 wt.% in dry gel). Significant differences, however, can be observed between shrinking factors, i.e., the ratios of swelling degrees at different temperatures e.g., at 25 and 50 °C, as also can be deduced from Figure 8 and Appendix A. These factors range between 15 (2B-5H) and 5 (4B) (see also kinetics discussion). Their value is reduced by rising crosslinker content (shorter elastic chains), while on the other hand, the content of HPMC present during synthesis raises the shrinking factor, due to looser and more regular structure of the network, as generated by the intercalation effect.

#### Kinetics of T-Response

The studied PNIPAm/HPMC hydrogels display fast to ultra-fast rates of response to the temperature jump from 25 to 50 °C, with shrinking times ranging down to 30 s. The shrinking times can be widely tuned (from 30 s up to several h) by the gels’ composition. Reswelling always was slow, requiring ca. 1 day. Intercalation of HPMC macromolecules into the PNIPAm network, and even more the micro-phase-separation of HPMC were found to accelerate the shrinking. The different rates of temperature-response correlate with the samples’ homogeneity, as well as with the gels’ different appearance during the deswelling process (see Figure 9).

The most homogeneous and slowest-shrinking gels display the formation of a stiff skin layer, on which large blisters (with thin skin) appear. They help the water expulsion from the gel (see 1B-1H in Figure 9a). This behavior is typical for simple chemically crosslinked monolitic PNIPAm gels [44]. In case of the still homogeneous but highly HPMC-intercalated sample 1B-5H (Figure 9b), the blisters are formed but are small and very numerous, while a fairly high shrinking rate is achieved: Just 8 min are needed for 73% of the deswelling, which is comparable with the fastest deswelling non-porous PNIPAm/starch gels studied in [25,26]. The cloudy but still semi-transparent sample 4B-1H (Figure 9c) displays no more blister formation during deswelling (at the 25→50 °C jump), but its shrinking rate is slow, similarly like in the case of 1B-1H. The nano-voids, left behind by the extracted HPMC and partly filled by the HPMC residue, obviously make possible a smooth albeit slow water transport through the ‘skin’ layer of the shrinking gel, so that no blisters are formed any more. The highly turbid gel 4B-5H (Figure 9d), which displays the highest micro-phase-separation among the studied gels, also keeps its smooth shape during shrinking, and it additionally achieves an ultra-fast shrinking rate: 30 s are needed for 78% of the deswelling process, which is comparable with ultra-fast-responsive super-porous nanocomposite PNIPAm gels (like in [13]), and which is nearly one order faster than the response of the fast-shrinking PNIPAm/starch gels in [25,26].

The graphs of the shrinking kinetics, triggered by the temperature jump 25→50 °C, are compared for all the studied gels in Figure 10 and also in Appendix A (2B series):

The curves in Figure 10 express the content of releasable water in % vs. time. It can be seen that the kinetics are accelerated by the increasing content of HPMC present during the synthesis (which subsequently was nearly completely extracted), and especially strongly by the micro-phase-separation during the synthesis, which leaves behind nano-pores in the place of the HPMC-rich domains (hence: fastest kinetics were found for 4B-2H till 4B-5H). Upon a temperature jump from 25 to 37 °C, the gels display similar kinetics like for 25→50 °C, as is illustrated by selected examples in Appendix A.

The phase separation occurs more readily at high crosslinker contents, especially at 4 mol% of BAA (series “4B”): all the 4B/HPMC gels except 4B-1H are ultra-fast responsive, similarly like 4B-5H. They hence combine excellent mechanical and tensile properties with an ultra-fast shrinking rate. This is especially true for 4B-3H. In case of the 2B series (see Appendix A), the micro-phase separation is more gradual, and the shrinking rates are more differentiated: response times can be easily tuned between 2 min and 2 h.

The HPMC-free gels based on pure PNIPAm, namely 1B, 2B, and 4B, display a ‘quenched deswelling’: after an initial moderately fast course, the process is blocked by a very strong skin effect. In contrast to that, such a quenching never is observed in case of the HPMC-intercalated gels, even if they shrink slowly. It should be also noted, that the quenching of the deswelling in 1B, 2B, and 4B did not occur during the stepwise temperature changes performed for recording the further-above-discussed temperature-dependent curves of swelling (*Q* = *f*(*T*) in Figure 8 and Appendix A).

The mechanism of the accelerated water release supported by the intercalated HPMC phase is illustrated in Figure 5a. Nano-pores loosely filled by the permanently fixed HPMC serve as channels for transporting water out of the gel. They at the same time are responsible for inhomogeneities in the refraction index, which lead to turbidity or opacity. In Figure 5b, the mechanism is shown on molecular level: Above the LCST temperature, the hydrogen bonds between water and PNIPAm are severed, PNIPAm contracts to a more compactly coiled conformation, while water is squeezed out along the PNIPAm/HPMC interface and further out through eventual nano-pores. The molecular mechanism was discussed in detail by the authors in [25,26] for PNIPAm/starch hydrogels, where the starch phase was not extractable, and where the maximum shrinking rates were slower (the shortest response times were 4–5 min), obviously due to the absence of the nanopores which are present in the presently studied PNIPAm/HPMC gels.

The gels in the as-prepared (“ap”) state lose most of their HPMC content during swelling experiments, i.e., during the equilibrium swelling at room temperature, as it was mentioned in the discussion of the synthesis. Nevertheless, the shrinking of one of such gels, 4B-2H-ap, was tested in response to the temperature jump from 25 to 50 °C, where the swelling degree changed between 9.7 (“ap” state) and 1.8 (fully shrunken). The kinetics, whose curve is shown in Figure 11 demonstrates, that the response time (2 min for 78% of the process) is comparable but slower than in case of the previously extracted and equilibrium-swollen sample (0.5 min/78%). In case of the sample 4B-2H-ap, not only water, but also much of the intercalated HPMC is ejected during the shrinking process. The situation in 4B-2H-ap is somewhat similar to the one in the PNIPAm/starch gels in [25,26] (with response time of 4–5 min).

### 2.6. Drug-Release Kinetics

An attractive potential application of the studied thermo-responsive hydrogels, especially of the ones with response times between ten and several tens of minutes, would be the temperature-triggered medicament- (or reagent) release, e.g., from a gel piece placed into the stomach. The fast-responsive gels, on the other hand, would be more attractive in the function of quasi-one-way actuators (rapid shrinking/fairly slow re-swelling).

In order to evaluate the drug release application, the kinetics of the release of the UV-absorbing drug Theophylline (see Figure 12a) was followed by means of UV/Vis spectroscopy (see Figure 12b–d). The fast-shrinking sample 4B-2H, as well as the slower-shrinking ones 1B-5H (transparent) and 4B-1H (cloudy) were selected for the experiments.

For loading the drug, the tested gels were first taken in the equilibrium-swollen and extracted state, then shrunken at 50 °C, and subsequently re-swollen (impregnated) in a Theophylline solution, as described in the Experimental Part and in the Appendix A.

The drug was released by putting the specimens into a bath of defined volume (distilled water), at the physiological temperature of 37 °C, at neutral pH. Solution samples were taken from the bath after pre-defined drug-release times, and were subsequently subjected to UV/vis spectroscopic analysis: The absorbance at 272 nm was measured as shown in Figure 12b and in Appendix A (the latter in the Appendix A, where also the raw data are listed and evaluated, see Appendix A). The evaluation of the release-time-dependent Theophylline concentration, under consideration of the removed solution volumes, yielded the kinetics of the release of this UV-absorbing drug. The results are analyzed in Figure 12c,d.

The degree of completion of the drug release process vs. time is plotted for the three selected specimens in the Figure 12c. The Figure 12d illustrates the simple deswelling process of the same three specimens (equally at 37 °C), in a separate experiment without the drug, at the same scaling like the kinetics of Theophylline release: it plots the degree of process completion vs. shrinking time. It can be seen, that within the error margin, the drug release and the shrinking of the ultra-fast deswelling sample 4B-2H follows the same kinetics (ca. 60% done in 1 min, ca. 90% done in 2 min). In case of the slower-shrinking samples 1B-5H (homogeneous, somewhat faster) and 4B-1H (cloudy, somewhat slower), the drug release times are the following: In case of 1B-5H, 60 min are needed for 68% of drug release, and 3 h for 90% release. In case of 4B-1H, 100 min are needed for 64% release, and 3 h for 76% release. A comparison of Figure 12c vs. Figure 12d shows, that the simple release of water is ca. two times slower than the release of Theophylline for both the slower-shrinking gels. This interesting difference can be explained by the fact, that the release of the drug is facilitated by the nano-channels present in the gels, which were left behind by the extracted HPMC phase. To sum up, it can be concluded, that among the studied products, the moderately fast shrinking hydrogels like 1B-5H and 4B-1H present promising, strong and tough materials able of drug release.

The above results, where non-porous hydrogels displayed in some cases exceptionally fast shrinking as well as drug release, are in an interesting contrast with some literature reports, e.g., with a work concerned with chemically related but highly porous hydrogels [34]. In the mentioned literature work, the drug release kinetics was comparable with our slower-responding gels in Figure 12c, while a simple expectation would be, that the porous gel from [34] should display the fastest shrinking, as well as the fastest drug release. A possible reason for the slow drug release from the porous gel in [34] might consist in a strong adsorption of the tested drug onto the polymer, and/or eventually in the presence of closed cell-porosity (which seems less likely because the material in [34] was described as aerogel).

## 3. Conclusions

Mechanically strong and extensible monolitic non-porous hydrogels based on divinyl-crosslinked poly(*N*-isopropylacrylamide) (PNIPAm) intercalated with linear hydroxypropyl methylcellulose (HPMC) were successfully developed. Morover, several of the gels additionally demonstrated ultra-fast response to temperature changes, which is exceptional for large non-porous gel specimens.

All the HPMC-intercalated PNIPAm hydrogels displayed a greatly improved extensibility and toughness.

The intercalation of HPMC during synthesis enforced the formation of a highly regular, and thus strong and extensible PNIPAm network. HPMC was practically quantitatively extracted after completed polymerization, thus yielding the final product, a ‘template-modified’ PNIPAm network intercalated with just a modest amount of permanently fixed HPMC.

Higher contents of divinyl crosslinker (2 or 4 mol%) as well as of the intercalated HPMC caused a varying degree of sub-micronic phase-separation in some of the products, which manifested itself first as blue opalescence, later as clouding or turbidity.

If the micro-phase-separation reached the degree of turbidity, a fast or even an ultra-fast T-induced shrinking was achieved near LCST of PNIPAm at 32 °C. The rate could be tuned by the degree of heterogeneity. The micro-phase-separation did not significantly deteriorate the moduli of the gels or their tensile properties.

From the practical point of view, the fastest-responsive among our hydrogels (shrinking time 30 s) could be of interest as soft actuators with one-way response (the re-swelling always was slow) and with very good mechanical properties, e.g., in soft robotics. The gels with moderately fast shrinking response (10 to several tens of minutes) in turn could be attractive for temperature-triggered drug delivery systems (as tested in this work with Theophylline), or for reagent release, as well as for related biomedical engineering applications.

## 4. Materials and Methods

### 4.1. Materials

*N*-isopropylacrylamide 97%, (NIPAm), *N*,*N′*-methylenebisacrylamide 99% (BAA), *N*,*N*,*N′*,*N′*-tetramethylethylenediamine ≥ 99.5% (TEMED) and ammonium peroxodisulfate reagent grade, 98% (APS) were purchased from Sigma-Aldrich (Burlington, MA, USA) and used as received. (Hydroxypropyl)methyl cellulose (HPMC) with the hydroxypropyl functionalization degree of 8.6% was purchased from Advent Chembio PVT LTD, Mumbai, India, and was used as received.

#### Synthesis

In a 20 mL cylindrical glass tube, the desired amount of HPMC was mixed with distilled water (reaction solvent, amount see Appendix A) at 25 °C, and the mixture was stirred for 24 h, so that a homogeneous solution was obtained. Subsequently, the monomer NIPAm and the divinyl crosslinker BAA were added, which was followed by thorough stirring. Next, the mixture was cooled down to 15 °C and purged with argon. Subsequently, TEMED, and thereafter the cooled 1% APS solution in water, were added (to start the radical polymerization) and briefly but thoroughly stirred under argon purging. Next, the mixture was immediately transferred into a 5 mL glass tube of 10 mm diameter (for obtaining a large specimen), as well as to several 5 mm tubes (NMR tubes, internal diameter 4 mm, used for obtaining additional thin specimens for tensile tests)—all these tubes were purged with argon prior to filling. After filling, the glass tubes were sealed, and the polymerization was left to proceed for 24 h at 25 °C. The so-obtained hydrogel products were taken out from the glass tubes (via cutting the tube and pushing out the sample). Next, some characterizations were carried out (moduli, tensile tests, optical microscopy) in the ‘as-prepared-state’ (some of the specimens were stored in this state). For obtaining standard specimens, the ‘as-prepared’ ones were immersed in distilled water at room temperature, and sol was extracted during 7 days, and the extraction bath was changed every day. In some cases, for determining the sol fraction, the water was collected, left to evaporate, and the sol was dried in air at 60 °C until constant weight which was recorded (ca. 5 days). Finally, the large cylindrical gel specimens were cut to pieces of approximately 10 mm height, the thin ones were cut to 15 mm, and were stored in water at room temperature, awaiting further use.

Stoichiometric ratios of the components: The employed concentration of monomer double bonds was 0.75 mol/L (corresponding to ca. 8 wt.% of the monomers NIPAm and BAA in the reaction mixture), and it was always kept constant, as well as the ratio APS/monomer double bonds (=0.0087), and the molecular ratio TEMED/APS (=3). The hydrogels were chemically crosslinked by different amounts of BAA: 1, 2, or 4 mol% of all C=C groups were from BAA. The amounts of components used for obtaining the different studied samples are given in Appendix A. The sample labeling in Appendix A is illustrated on the following example: “4B-1H” means a sample, which contains the 4 mol% of C=C from BAA, relatively to all polymerizable double bonds, and 1 wt.% of HPMC in the reaction mixture. As can be seen in Appendix A, the amount of HPMC in dry product is ca. 10 times higher than in the ‘as-prepared’ hydrogel: e.g., 2B-1H contains 10.7 wt.% of HPMC, while the ‘filler’ content in ‘as-prepared’ 2B-5H is 38.4 wt.%.

### 4.2. Methods

#### 4.2.1. Micron-Scale Morphology

The micron-scale morphology and the transparency of the non-porous monolithic hydrogels was studied using the DM6000 M light microscope from Leica Camera AG (Wetzlar, Germany). The images were taken in the transmitted-light-mode.

#### 4.2.2. Nano-Morphology Studied by Transmission Electron Microscopy (TEM)

The morphology of the gels was examined by means of TEM, on the microscope Tecnai G2 Spirit Twin 12 from FEI (Brno, Czech Republic). In order to prepare the gels for observation without altering their morphology (e.g., by pore generation during freeze-drying), the selected specimens were first left to slowly dry in air at ambient temperature (for ca. 3 days), and subsequently, the drying of the glassy samples was continued at 60 °C for 5 days. The dry glassy non-porous specimens were subsequently cut to ultrathin sections (specimens) using an ultramicrotome (Ultracut UCT, from Leica, Wetzlar, Germany). The ultrathin sections were transferred onto a microscopic grid covered with a thin carbon film, in order to improve their stability under the electron beam during TEM observations.

#### 4.2.3. FTIR Spectroscopy

FTIR spectra of the dried hydrogel samples were recorded on a Nicolet Nexus 870 FTIR spectrometer (from Thermo Scientific, Madison, WI, USA, now Thermo Fisher Scientific, Waltham, MA, USA), using an ATR—Specac MKII Golden Gate Single Reflection ATR System with a diamond crystal, and with the angle of incidence of 45°. An MCT detector was used. The resolution was 4 cm^−1^, the number of scans per spectrum: 256.

#### 4.2.4. Shear Moduli of the Hydrogels

Shear moduli (*G*) of gels after synthesis, as well as of swollen gels were measured between parallel plates in slow uniaxial compression mode at room temperature using an ARES-G2 machine (a multi-functional DMTA/rheometer) from TA Instruments, New Castle, DE, USA. The compression was gradually increased from 0% to 10% in 2 min, while the applied compression force was recorded. The shear modulus *G* was calculated according to [43] using the Equation (1):*G* = *F*/*S*_0_ (*λ*^−2^ − *λ*)(1)
where:

*F* is the compression force, *S*_0_ is the initial cross-section of the sample before measurement, *λ* = *l*/*l*_0_ is the relative deformation, while *l* and *l*_0_ are the compressed and the initial heights of the sample, respectively. Moduli were determined as average values from three experiments.

#### 4.2.5. Tensile Tests

For the tensile tests all the hydrogel samples were prepared in glass tubes with internal of 5 mm. The tensile properties of small and thin samples of the prepared gels were measured using an ARES-G2 machine (a multi-functional DMTA/rheometer) from TA Instruments, New Castle, DE, USA (maximum allowed force with this machine: 20 N), at room temperature, with a cross-head speed of 0.25 mm/s. Thin cylindrical samples with the geometry: total specimen length: 15 mm, length between jaws: 3 mm and diameter of 4 mm were used. At least three measurements were carried out for each sample. Presented are the tensile curves closest to the average one.

#### 4.2.6. Temperature Dependence of the Swelling Degree in Water

The swelling degree *Q* of the hydrogel samples was determined as: *Q* = *m_sw_*/*m_dry_*. where: *m_sw_* is the mass of the swollen gel, *m_dry_* is the mass of the dry gel (this latter was determined after finishing all swelling tests with the given specimen). For drying the specimen, it was first dried at room temperature in air for 24 h, and thereafter in vacuum at 100 °C until weight constancy (typically 24–48 h).

For obtaining the temperature dependence of *Q*, the gel specimen was kept in the water bath at the given temperature for 24 h. Thereafter, the specimen was weighed. The *Q* values were determined at *T* = 22 °C, and subsequently at higher temperatures up to *T* = 50 °C with a step of 2 °C.

#### 4.2.7. Deswelling ‘Kinetics’ Triggered by T

The kinetics curves (*Q* = *f*(*time*) plots) were determined like the above *Q* = *f*(*T*) curves. An equilibrium-swollen sample was transferred into a bath with a temperature of 50 °C in order to generate the stimulus. The weight change in time was subsequently recorded.

The relative content of releasable water in the gels was calculated using the Equations (2)–(5), from the time-dependent degree of swelling and expressed in % of the amount of swelling water which can be released, and which is present in the sample at a given time point. The starting point (here equilibrium swollen at 25 °C) corresponds to 100%; the final point, equilibrium shrunken (here at 50 °C), corresponds to 0%. The calculation was performed as follows:(2)$$content\;of\;releasable\;water=100\%\;(1-\frac{\Delta d_{i}}{\Delta d_{max}})$$
where:(3)$$relative\;swelling\;change=100\%\;\frac{\Delta d_{i}}{\Delta d_{max}}$$
(4)$$\Delta d_{i}=Q_{max}-Q_{i}$$
(5)$$\Delta d_{max}=Q_{max}-Q_{min}$$
and *Q_i_* is the actual degree of swelling, *Q_min_* is the degree of swelling at 50 °C and *Q_max_* is the equilibrium degree of swelling at 25 °C.

#### 4.2.8. Drug Release Kinetics

Specimens of tree selected gels were tested, namely of 1B-5H, 4B-1H and 4B-2H. Each specimen was impregnated for the drug delivery test as follows: A cylindrical specimen of ca. 1 g weight (in the equilibrium swollen state at 25 °C) was taken and subjected to deswelling in hot distilled water (50 °C) for 24 h. Subsequently, the mass of the shrunken specimen was recorded and it was put into 15 mL of a (concentrated) solution of the tested drug to be released, Theophylline (*c_impregnation_* = ca. 0.036 mol/L), where the specimen underwent re-swelling (drug-impregnation) for 48 h. Thereafter, the re-swollen mass was recorded in order to later calculate the absorbed amount of the drug (and to verify the completeness of re-swelling).

For measuring the drug-release kinetics, the drug-impregnated (‘loaded’) specimen was put into a ‘release bath’ of precisely 250 mL, consisting of distilled water (pH = 7) which had a temperature of 37 °C, in order to generate the triggering stimulus. Thereafter, several 3 mL samples were taken after suitable release times (which were accurately recorded). The removed probe volumes also were precisely noted, and were considered in the experiment evaluation. Next, the concentrations of Theophylline in the probes taken at specific process times were determined using UV/vis spectroscopy. From the determined concentrations (while considering the small removed probe volumes), the amounts of Theophylline released at given time points were calculated in several steps. A detailed description of the data evaluation is given in the Appendix A, accompanied by tables with experimental data and the magnitudes derived from them. The drug-release kinetics experiment was repeated two times for each tested hydrogel composition.

#### 4.2.9. UV/Vis Spectroscopy

Ultraviolet-visible (UV/vis) spectra were measured on a Perkin-Elmer “Lambda 35 UV/VIS” spectrometer from PerkinElmer (Waltham, MA, USA). 3 mL samples of the analyzed solutions were measured in 10 mm thick quartz cuvettes. The UV/vis experiments were carried out at room temperature.

## Data Availability

The data presented in this study are openly available in article.

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
