# Peer review of "Exceptionally Fast Temperature-Responsive, Mechanically Strong and Extensible Monolithic Non-Porous Hydrogels: Poly(N-isopropylacrylamide) Intercalated with Hydroxypropyl Methylcellulose"

_gels, 2023, doi:10.3390/gels9120926_

Round 1

Reviewer 1 Report

Comments and Suggestions for Authors

The shear modulus G needs more explanation. As the symbols in the equation are not clarified. 

line 263 needs to be revised.

Is the prepared gel regenerated to its original after drying?

What is the economic income from this study?

Comments on the Quality of English Language

Language typing eg. line 263, also the symbols of equations line 222

Author Response

Response to evaluation table:

 The authors are very grateful for the generally positive assessment in the above table, as well as for the attentive and critical reading of the paper, and for the valuable questions and suggestions which helped to improve the Manuscript and make it more attractive and reader-friendly.

The changes and added texts in the revised Manuscript, which were done/added in response to comments of Reviewer #1, are highlighted with yellow color in the uploaded “PDF version with highlighted changes”.

Reviewer #1:#

The shear modulus G needs more explanation. As the symbols in the equation are not clarified.

Response:

The original text (lines 220 to 225 in the submitted Manuscript) indeed was not reader-friendly. Now, in the revised Manuscript (lines 223 to 230), the text is improved and all the magnitudes used for the calculation of G are explained.

Reviewer #1:

line 263 needs to be revised.

Response:

The sentence on line 263 was improved in order to be more reader-friendly (now line 269 in the revised Manuscript). The introducing paragraph, of which the mentioned sentence was part, was joined with the following paragraph, as response to comments of Reviewers #2 and #4.

Reviewer #1:

Is the prepared gel regenerated to its original after drying?

Response:

Yes, but several days are needed for regenerating a large bulk specimen. In case of extremely swelling samples, which were not studied in this work (our present samples were gently swelling), a gradual re-swelling procedure would be necessary, in order to prevent cracking of the re-swelling samples. The possible regeneration is mentioned in the revised Manuscript on lines 449 to 452.

The authors are grateful for the interest in this specific topic. The authors consider the topic of gentle (slow) drying vs. freeze drying to be very important, as the freeze-drying automatically generates a porous morphology (whose pattern depends on the freeze-drying conditions).

Reviewer #1:

What is the economic income from this study?

Response:

The authors perceive an application potential in the fields of soft actuators (and hence soft robotics), in drug delivery systems, as well as in related biomedical engineering applications. This is now better stressed in the Abstract and in the Conclusions of the revised Manuscript.

Reviewer #1:    Comments on the Quality of English Language:

Language typing eg. line 263, also the symbols of equations line 222   

Response:

The authors checked the whole Manuscript for reader-friendliness (in addition to lines 222 and 263). The extensively improved text fragments are highlighted by color, but the numerous ‘cosmetic’ changes which resulted from this check are not highlighted. 

Reviewer 2 Report

Comments and Suggestions for Authors This manuscript describes the fabrication and characterization of hydroxypropyl methylcellulose (HPMC) intercalated divinyl-crosslinked poly(N-isopropylacrylamide) (PNIPAm) non-porous hydrogels, which displayed a greatly improved extensibility and toughness, and ultra-fast response to temperature changes. However, I think there are some points that need to be clarified and more carefully explained: 1.Please delete the phrases or sentences with bold font in whole manuscript, and the author should re-organize the paragraph only comprising of one sentence. 2.In literature [34], it reported the semi-interpenetrating PNIPAm-based networks with highly porous. Why does the similar system herein have inconsistent phenomenon with non-porous? Where is difference? 3.In Figure 1, the unit in Y caption is missing. 4.In Figure 9 and 10, the uppercase of first word in X and Y caption should be kept a consistent style.

Comments on the Quality of English Language

Minor editing of English language required

Author Response

Reviewer #2:

This manuscript describes the fabrication and characterization of hydroxypropyl methylcellulose (HPMC) intercalated divinyl-crosslinked poly(N-isopropylacrylamide) (PNIPAm) non-porous hydrogels, which displayed a greatly improved extensibility and toughness, and ultra-fast response to temperature changes. However, I think there are some points that need to be clarified and more carefully explained:

Response:

The authors are very grateful for the generally very positive assessment in the above table and in the comments, as well as for the attentive and critical reading of the paper, and for the valuable questions and suggestions which helped to improve the Manuscript and make it more attractive and reader-friendly.

The changes and added texts in the revised Manuscript, which were done/added in response to comments of Reviewer #2, are highlighted with green color in the uploaded “PDF version with highlighted changes”.

Reviewer #2:

1.Please delete the phrases or sentences with bold font in whole manuscript, and the author should re-organize the paragraph only comprising of one sentence.

Response:

The authors removed the mentioned bold formatting and merged (or moved and merged) the paragraphs consisting of one sentence. Some starting sentences of the paragraphs were revised in order to give a better orientation in the first words. Important changes related to this point (which was also raised by Reviewer #4) are highlighted in magenta color.

Reviewer #2:

2.In literature [34], it reported the semi-interpenetrating PNIPAm-based networks with highly porous. Why does the similar system herein have inconsistent phenomenon with non-porous? Where is difference?

Response:

This is indeed an interesting question. In the citation [34] a highly porous material was prepared by means of freeze drying of a previously non-porous semi-interpenetrating PNIPAm-based network. Theoretically, the shrinking, and also the drug release should be faster from the porous material, especially in comparison with the non-porous and large specimens. Possible reasons for the slow response of the porous material in [34] might be a specific strong adsorption of the tested drug onto the polymer (in [34]), and/or eventually (which the authors consider less probable) the presence of closed cell-porosity (as observed in the authors work [13]).

Reviewer #2:

3.In Figure 1, the unit in Y caption is missing.   

Response:

The unit in Y caption of Figure 1 (Swelling degree) was dimensionless, and originally written as “( )”. In the revised Manuscript it was corrected to “(-)”, for the sake of improving clarity. All the graphs in the Manuscript and in the Supplementary Information File were checked, and eventually corrected accordingly.

Reviewer #2:

4.In Figure 9 and 10, the uppercase of first word in X and Y caption should be kept a consistent style.     

Response:

The X and Y captions in the mentioned Figures were improved as suggested. All the graphs in the Manuscript and in the Supplementary Information File were checked, and eventually corrected accordingly.

Reviewer #2: Comments on the Quality of English Language:

Minor editing of English language required.

Response:

The authors checked the whole Manuscript for reader-friendliness. The numerous ‘cosmetic’ changes which resulted from this check are not highlighted by color.

Reviewer 3 Report

Comments and Suggestions for Authors

Author Response

Reviewer #3:

I am writing to express my enthusiasm and support for the acceptance of the paper titled "Exceptionally fast temperature-responsive, mechanically strong and extensible monolithic non-porous hydrogels: poly(N-isopropylacrylamide) intercalated with hydroxypropyl methylcellulose" Authored by Beata Strachota et al. the paper delves into the innovative synthesis of hydrogels that exhibit remarkable characteristics.

The research work is exceptionally compelling, primarily focusing on the synthesis of hydrogels with extraordinary mechanical strength, rapid temperature responsiveness (achieving a remarkable 30-second response time), and noteworthy extensibility. The intercalation effect within the PNIPAm network, coupled with the toughening through hydrogen bonding with HPMC, presents a unique structural configuration that enhances the hydrogel's performance.

I believe that the findings presented in this paper hold substantial promise for various fields, including but not limited to materials science, biomedical engineering, and drug delivery systems. The depth of the study and the clarity of its presentation significantly contribute to the advancement of knowledge in this domain.

Therefore, I strongly recommend the acceptance of this paper for publication in your esteemed journal. The comprehensive insights and the innovative approach demonstrated in this research will undoubtedly captivate the interest of your readership and contribute significantly to the ongoing discourse in the field of hydrogel research.

And this article is well written, logical, well-illustrated and contains many interesting facts. The topic of research is relevant and ...

Response:   The authors are very grateful for the very positive assessment, as well as for the interest of the Reviewer in our topic, and for the below suggestions which helped to improve the Manuscript.

The changes and added texts in the revised Manuscript, which were done/added in response to comments of Reviewer #3, are highlighted with grey color in the uploaded “PDF version with highlighted changes”.

Reviewer #3:

I recommended acceptance after addressing minor comments:

- (Disclaimer/Publisher’s Note) put this paragraph after references.

- Author Contributions:/

- Funding:/

- Informed Consent Statement:/

- Data Availability Statement:/

- Conflicts of Interest:/

fill these parts.

Response:

The mentioned information was fulfilled and moved (in case of the disclaimer statement).

Reviewer 4 Report

Comments and Suggestions for Authors

This manuscript has the potential to be approved and published in Gel, provided that the authors adequately address the following inquiries and suggestions:

1. Within the article section, numerous words or sentences are rendered in bold font to convey emphasis. However, it is my contention that such a practice is unnecessary, as the text section should ideally maintain a consistent and uniform style throughout.

2. There exist paragraphs within the text that are comprised of a single sentence, necessitating more expansion and development..

3. Given the absence of limitations on the inclusion of tables or figures within this publication, it is advisable to incorporate supplementary information directly into the article, thereby facilitating readers' access to pertinent data. 

4. Within the synthesis part, the author presents a sub-section entitled "comment". What is the intended objective of this sub-section?

5. All the equations should be numbered.

6. For drug release kinetics experiments, what pH value is used?

7. How many independent measurements are taken for a drug release kinetics experiment? The error bars should be shown in Figure 9. 

Author Response

Reviewer #4:

This manuscript has the potential to be approved and published in Gels, provided that the authors adequately address the following inquiries and suggestions:

Response:

The authors are very grateful for the very positive assessment in the above table and in the comments, as well as for the attentive and critical reading of the paper, and for the valuable questions and suggestions which helped to improve the Manuscript and make it more attractive and reader-friendly.

The changes and added texts in the revised Manuscript, which were done/added in response to comments of Reviewer #4, are highlighted with turquoise color in the uploaded “PDF version with highlighted changes”.

Reviewer #4:

  1. Within the article section, numerous words or sentences are rendered in bold font to convey emphasis. However, it is my contention that such a practice is unnecessary, as the text section should ideally maintain a consistent and uniform style throughout.
  2. There exist paragraphs within the text that are comprised of a single sentence, necessitating more expansion and development.

Response:

The authors removed the mentioned bold formatting and merged (or moved and merged) the paragraphs consisting of one sentence. Some starting sentences of the paragraphs were revised in order to give a better orientation in the first words. Important changes related to this point (which was also raised by Reviewer #2) are highlighted in magenta color.

Reviewer #4:

  1. Given the absence of limitations on the inclusion of tables or figures within this publication, it is advisable to incorporate supplementary information directly into the article, thereby facilitating readers' access to pertinent data.

Response:

The authors are very grateful for this comment, as they usually (and regularly) receive opposite suggestions concerning the placement of Figures and Tables. In response to this comment, three most interesting items were moved from the Supplementary File into the Manuscript, namely: new Scheme 2 in the revised Manuscript (ex Scheme S1), new Table 1 (ex Table S3), new Figure 7 (formerly part of the larger Figure S7) and new Figure 11 (ex Figure S11).

Reviewer #4:

  1. Within the synthesis part, the author presents a sub-section entitled "comment". What is the intended objective of this sub-section?

Response:

The sub-section name was corrected to “Stoichiometric ratios of the components”, which is much more descriptive.

Reviewer #4:

  1. All the equations should be numbered.

Response:

This was corrected as suggested.

Reviewer #4:

  1. For drug release kinetics experiments, what pH value is used?

Response:

The drug release kinetics were conducted in neutral pure (distilled) water, at pH = 7. This is now better stressed in the Discussion. (according to our experience, an acidic pH would likely moderately accelerate the release kinetics, due to the ionic effect of the buffer on PNIPAm.

Reviewer #4:

  1. How many independent measurements are taken for a drug release kinetics experiment? The error bars should be shown in Figure 9.

Response:

Two measurements were done per each drug release kinetics experiment. This is now mentioned in the revised Manuscript.  Error bars were added to Figure 9 and to other Figures where they were missing.

Round 2

Reviewer 4 Report

Comments and Suggestions for Authors

This manuscript can be accepted for publication